# Adhesion Behavior of Textured Electrosurgical Electrode in an Electric Cutting Process

**Caiying Zhou, Juncheng Lu and Xingsheng Wang \***

College of Engineering, Nanjing Agricultural University, Nanjing 210031, China;
zhoucaiying1223@163.com (C.Z.); JunchengLu1998@163.com (J.L.)

**\*** Correspondence: xingshengwang@njau.edu.cn

**Abstract:** Soft tissue adhesion on the electrosurgical electrode has been a major concern in clinical surgery. In order to improve the adhesion property of the electrode, micro-textures with different morphologies including micro-dimples, longitudinal micro-channels, and lateral micro-channels were created on the electrode surface by laser surface texturing (LST). Electric cutting experiments were then performed to investigate the adhesion behavior of different electrodes. Experimental results showed that the textured electrode surfaces could reduce the soft tissue adhesion significantly due to the effect of air in micro-textures and the reduction of contact area between the electrode and the soft tissue. Moreover, the temperature distribution of the electric cutting process was simulated through COMSOL to verify the effect of different micro-textures on adhesion behavior. It was demonstrated that the better anti-adhesion property could be obtained at a large area density combined with lateral micro-channels.

**Keywords:** anti-adhesion; micro-texture; laser surface texturing; electrosurgical electrode; temperature distribution

## 1. Introduction

The application of an electrosurgical electrode instead of the traditional scalpel in clinical surgery reduces pain and trauma, and improves recovery of the patients [1,2]. Nevertheless, the electrosurgical electrode can reach a very high temperature during the operation [3], accumulating heat in a short time, which often results in some clinical events, such as charring scab, biofouling, and carbonized eschar, etc. [4]. Even worse, high-frequency electric field can induce more tight tissue adhesion on the electrosurgical electrode surface than the single thermal field [5], making it difficult to clean off the soft tissue adhering to the electrode. During the operation, the above problems are solved by removing the adherent tissue on the electrode or frequently replacing the electrodes. This extends the operation time, and affects the entire operation process [6]. Therefore, it is necessary to improve the anti-adhesion performance of the electrosurgical electrode.

In recent years, some studies have been carried out to reduce the soft tissue adhesion on the electrosurgical electrode surfaces. For example, Dodde et al. [7] added a cooling channel filled with fluid coolant on the electrode to increase the cooling rate of soft tissue, thus achieving a reduction of the thermal damage of soft tissue. Donzelli et al. [8] used cooling water to irrigate the tip of the electrosurgical electrode, leading to a reduction of thermal damage and related tissue adhesion. Morales et al. [9] irrigated a surgery site with cold saline at a certain rate to reduce thermal damage and tissue adhesion during the operation. However, these cooling methods require auxiliary devices, which increases the difficulty and cost of processing. Some studies have tried to reduce tissue adhesion by depositing coatings on the electrosurgical electrode surface. For instance, Kang et al. [10] deposited hexamethmyldisiloxane (HMDSO) on the metallic jaws of surgical devices by employing

atmospheric pressure RF-driven plasma to create a non-stick coating. It was demonstrated that surface coatings significantly reduced the soft tissue adhesion on the instrument. The plasma was produced in a generator under the action of an electric field, but the process was unstable at a high power, making the whole machining process difficult to adjust and control. Shen et al. [11] proposed the RF magnetron-sputtering technique for preparing DLC-Cu (diamond-like carbon) thin films on the electrode surface to reduce tissue adhesion. Surface coatings can reduce the thermal damage and tissue adhesion effectively, but these coatings were usually unstable and easily damaged at a high temperature. Moreover, the preparation cost of metal coatings was high. In view of the drawbacks of the above methods, another feasible method is to fabricate micro-textures on the electrode surface.

Various surface textures have been applied to improve the adhesion properties of materials in recent years, such as reducing bacteria adhesion [12,13], tuning cell adhesion [14,15], reducing particle adhesion [16,17], improving anti-adhesion properties of metal surface [18] and cutting tools [19,20], etc. In general, a lot of textures are inspired by natural surface textures of living organisms. The created micro-textures are very close to the original morphologies of the biological surfaces in some studies. For example, Liang et al. [21] created a hierarchical micro-hole array combined with nano-structures on a titanium surface by imitating the multilevel structures of natural bones so that the cell adhesion was tuned successfully. Bhushan et al. [22] designed nano-structures on the micro-textured surfaces, thus creating a hierarchical structure with complex morphologies inspired by shark skin and lotus leaf. The machined surface showed the properties of a large contact angle and low adhesion after processing. However, considering the limitation of actual processing conditions and high costs related to these techniques, more simple micro-textures were designed, such as micro-channels, and micro-dimples. For example, Zhou et al. [23] fabricated micro-channels by laser surface texturing (LST), thus preparing bionic non-smooth surfaces on a mold. It was found that the created micro-textures improved anti-adhesion properties of the mold, and the adhesion properties were closely related to the area density of micro-textures. Laser induce periodic surface structures (LIPSS) were created by Orazi et al. [24] on stainless steel, aluminium and copper substrates, and the experimental results showed that LIPSS turned the metals from hydrophilicity to hydrophobicity as well as the adhesion property. Wang et al. [25,26] fabricated different micro/nano-hierarchical structures on stainless steel surface, thus leading to different wetting behaviors and adhesion properties. Sun et al. [27] prepared grid-like micro-textures on 45 steel by LST. The adhesion force between ceramic products and the textured molds decreased significantly. Simplified micro-textures were also applied to improve the anti-adhesion properties of electrosurgical electrodes. For example, Zhang et al. [28] utilized a micro-contact printing method for fabricating micro-dimples on the electrode surface. Electric cutting experiments results showed the soft tissue adhesion on the textured electrode obviously reduced. However, self-assembly deposition and electrochemical etching techniques were used for preparing anti-adhesion surfaces, which complicated the process. Similarly, inspired by a maize leaf, Han et al. [29,30] designed grid-like structures and prepared a $TiO_2$ coating on the electrode surface. Excellent anti-adhesion properties of the coupled bionic electrode were demonstrated in electric cutting experiments. In general, there are two main problems in designing and fabrication of micro-textures at present, i.e., the complexity of the machining process and a lack of comparison between different textures.

To solve the problems mentioned above, several different kinds of micro-textures (i.e., micro-dimples, longitudinal micro-channels and lateral micro-channels) used in previous studies for improving adhesion properties are created to find the micro-texture that could produce the most effective anti-adhesion properties. Due to the simple preparation process and high processing efficiency, LST is a feasible and effective method for fabricating micro-textures to improve surface properties [31,32]. Micro-textures with different morphologies and distinct area densities are created on the electrode surface by LST. Electric cutting experiments are performed to explore the effect of morphology and area density on adhesion properties. Meanwhile, the simulation based on COMSOL multiphysics is used to verify the influence of micro-textures with different morphologies on temperature distributions during the electric cutting process. This study presents a feasible approach to decrease soft tissue

adhesion on the electrode, and provides an effective method to improve anti-adhesion properties of other electrosurgical instruments.

## 2. Experimental Section

### 2.1. Fabrication of Micro-Textures

In this study, a Nd:YVO4 picosecond laser (PX50, Edgewave, Würselen, Germany) generating a pulse duration of 10 ps and a wavelength of 532 nm was utilized to fabricate micro-textures. The laser beam was focused on the electrode surface through the transmission system consisting of several optical mirrors, a half-wave plate, a liner polarizer and a focal lens, as shown in Figure 1. The resulting spot diameter was about 16 μm at the focal plane. The electrosurgical electrode was mounted on a 2-axis motion *x-y* stage (Planar100XY, Borui, Shanghai, China) with a translation precision of 500 nm which was controlled by a controller (PMAC, Delta, CA, USA) to achieve micro-textures machining. Furthermore, a charge coupled device (CCD) camera (acA2500-14gc, Schneider, Bart Kreuznach, Germany) was used to align the laser spot to the edge of the electrode, which was connected to the personal computer (PC) via a GigE Internet cable.

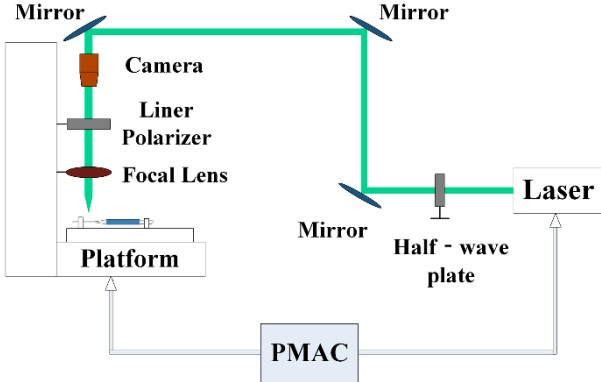

**Figure 1.** Schematic of laser surface texturing (LST).

The electrosurgical electrode made of 304 stainless steel with a length of 12 mm, a width of 2 mm, and a thickness of 0.5 mm was used in the whole experiments. Micro-textures with three kinds of different morphologies (i.e., micro-dimple, lateral micro-channel and longitudinal micro-channel) were fabricated on the electrode surface, respectively. The micro-dimples with a diameter of 50 μm and a depth of 12 μm were designed. A spiral scanning trajectory similar to a spiral shape was employed to fabricate micro-dimples. A pulse frequency of 10 kHz, a laser pulse energy of 0.4 μJ, and a scanning speed of 2 mm/s were used during LST. The center-to-center distance of the micro-dimples was 100 μm, thus leading to a micro-dimple array with an area density of 20%. Laser scanning was performed by using a raster trajectory to fabricate micro-channels. Longitudinal micro-channels were created when the laser beam was scanned in the longitudinal direction (parallel to the handle of the electrode), and lateral micro-channels were created in the lateral direction (perpendicular to the handle of the scalpel). The scanning speed and laser pulse frequency used to fabricate these two kinds of micro-channels were the same as that of fabricating micro-dimples. A laser pulse energy of 1.2 μJ was used for realizing machining. The width and depth of the created micro-channels were 50 and 12 μm, respectively. The center-to-center distance of the micro-channels was 250 μm. After LST, these three kinds of micro-textures with an area density of 20% can be obtained through different processing trajectories. The anti-adhesion properties of the three kinds of micro-textures were investigated under the area density of 20%. According to subsequent electric cutting experiments, the electrode with lateral micro-channels exhibited the best anti-adhesion property with soft tissue. In order to obtain better parameters for improving the adhesion behavior of electrode, lateral micro-channels with different

area densities (from 10% to 40%) were fabricated. The parameters of the designed micro-textures are shown in Table 1.

**Table 1.** Shape characteristics and geometric parameters of micro-textures on the electrode surface.

| Electrode No. | Width (μm) | Depth (μm) | Micro-Texture Characteristics | Area Density (%) |
|---|---|---|---|---|
| 1 | 0 | 0 | Without micro-texture | 0 |
| 2 | 50 | 12 | Micro-dimples | 20 |
| 3 | 50 | 12 | Longitudinal micro-channels | 20 |
| 4 | 50 | 12 | Lateral micro-channels | 20 |
| 5 | 25 | 12 | Lateral micro-channels | 10 |
| 6 | 75 | 12 | Lateral micro-channels | 30 |
| 7 | 100 | 12 | Lateral micro-channels | 40 |

The electrodes were cleaned with an ultrasonic cleaner (PS-D40A, Kangjie, Dongwan, China) to remove debris from the processed area after LST. Then, a 3D laser confocal scanning microscope (OLS-4100, Olympus, Tokyo, Japan) was used to observe the surface morphologies of the micro-textures. The depth, diameter, and width of six different micro-textures were averaged and recorded.

*2.2. Cutting Experiments*

In order to prevent large deviations of the experimental results caused by manual cutting, a designed setup comprising of a tissue clamp, a vertical lift table, a sheet clamp, a linear motor module, a two-axis horizontal movement platform was utilized for electric cutting experiments, as shown in Figure 2. The electric cutting experiments were performed by using a minimally invasive electrosurgical device (LK-3, JIUJIAN Medical Equipment, Shenzhen, China). During the electric cutting experiments, the linear motor module (MLCJ-0040-075-00, Junxin, Shanghai, China) controlled by a Copley drive (ACJ-055-18, Copley, MA, USA) was utilized to realize translational motion.

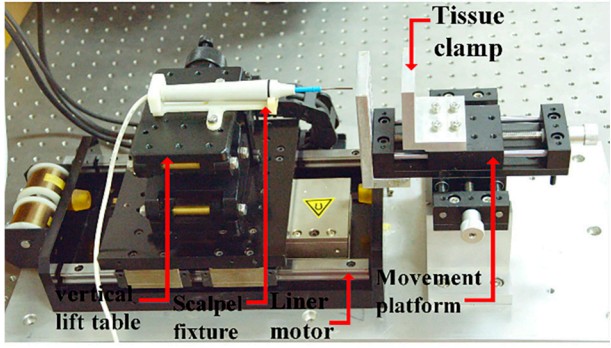

**Figure 2.** Experimental setup for the electric cutting experiment.

Based on the previous studies [5,28,29], fresh porcine liver bought from a local Warmart market was used in the whole electric cutting experiments. In order to prevent deformation and dislocation of soft porcine liver during the cutting process, the porcine liver was cut and put into a box with a size of $20 \times 20 \times 80$ mm$^3$ for compaction and fixation. The box was clamped and fixed on the movement platform by the tissue clamp.

Pure cutting mode of the minimally invasive electrosurgical device was adopted for electric cutting experiments. The electric cutting power was set as 32 W. The electrode was fixed on the longitudinal lift table by a fixture, and inserted horizontally into the liver tissue. The whole electric cutting experiments were performed under a cutting speed of 2.5 mm/s and a cutting time of 4 s, which resulted in an incision depth of around 10 mm. For every electrode, electric cutting experiments were repeatedly performed four times at different locations of the liver tissue under the same experimental condition. After every electric cutting experiment, an electronic analytical balance (PTY-2003, HZ&HUAZHI, IL,

USA) was used to weigh the adhesion mass of different electrodes. All the electric cutting experiments were performed at a constant temperature of approximately 24 °C, and completed within 3 h to prevent deterioration of the biological tissue. Experiments were repeated five times for every kind of electrode under the same condition.

### 2.3. Numerical Implementation

In order to further verify the effect of different micro-textures on the temperature field, finite element analysis were performed in COMSOL multiphysics to simulate the temperature distributions during the cutting process. Biological heat transfer coupled with a Joule thermal module was used for simulation. The biological tissue was identified as the porcine liver, and the electrode material was set as 304 stainless steel. According to the previous studies [7,33], the property parameters of the two materials are shown in Table 2.

**Table 2.** Electrical and thermal properties of materials.

| Finite Element Analysis (FEA) Region | Material | $c$ (J/kg·K) | $\rho$ (kg/m$^3$) | $k$ (W/m·K) | $\sigma$ (s/m) |
|---|---|---|---|---|---|
| Biological Tissue | Porcine Liver | 3600 | 1060 | 0.512 | 0.333 |
| Electrode | 304 stainless steel | 132 | 21,500 | 71 | $4 \times 10^6$ |

Pennes' biological heat transfer equation was adopted, and the thermodynamic equation was specified as follows [34]:

$$\rho c \frac{\partial T}{\partial t} + \nabla(-k\nabla T) = -Q_{\mathrm{p}} + Q_{\mathrm{met}} + Q_{\mathrm{ext}} \tag{1}$$

where $\rho$ is tissue density (kg/m$^3$), $c$ is heat capacity (J kg$^{-1}$·K$^{-1}$), $k$ is thermal conductivity (W m$^{-1}$·K$^{-1}$), $Q_{\mathrm{met}}$ is the metabolic heat production of biological tissue, $Q_{\mathrm{ext}}$ is the external heat production due to the tissue-heating electrode, and $Q_{\mathrm{p}}$ is blood perfusion rate. $Q_{\mathrm{met}}$ and $Q_{\mathrm{p}}$ could be ignored since the experiments were performed in vitro. In order to reduce the calculation time, all material properties were set as constants ignoring the variation with the temperature. The porcine liver and electrode were modeled according to the solid size. The adiabatic boundary was applied to the lower surface, while the electrical insulation boundary was applied to the side and upper surfaces of the liver model. The convection boundary conditions were applied on the upper surface of the liver tissue which was in contact with the electrode, and a convection coefficient $h = 25$ W m$^{-1}$·K$^{-1}$ was given according to reference [7]. A voltage of 50 V corresponding to 32 W output power was applied to the surgical electrode. The initial voltage of the ground plate that contacts with the soft tissue was set to 0 V. The activation time was set to 4 s. Considering the limitation of the computing resources, the static process was simulated instead of the dynamic cutting process.

## 3. Results and Discussion

### 3.1. Morphologies of the Textured Electrode Surface

Three kinds of different micro-textures with an area density of 20% are shown in Figure 3a–c. Obviously, their distributions on the electrode surface are different under the same area density. The 3D enlarged images of micro-dimples, longitudinal micro-channels and lateral micro-channels are shown in Figure 3d–f, respectively. Different micro-morphologies of the three kinds of texture can be seen clearly. The comparison of cross-sectional profiles of these micro-textures is shown in Figure 3g. As can be seen, the sectional dimensions of all three kinds of micro-textures are almost the same. The width of micro-channels is 50 μm, and the depth is approximately 12 μm. The diameter and depth of micro-dimples are 50 and 12 μm, respectively.

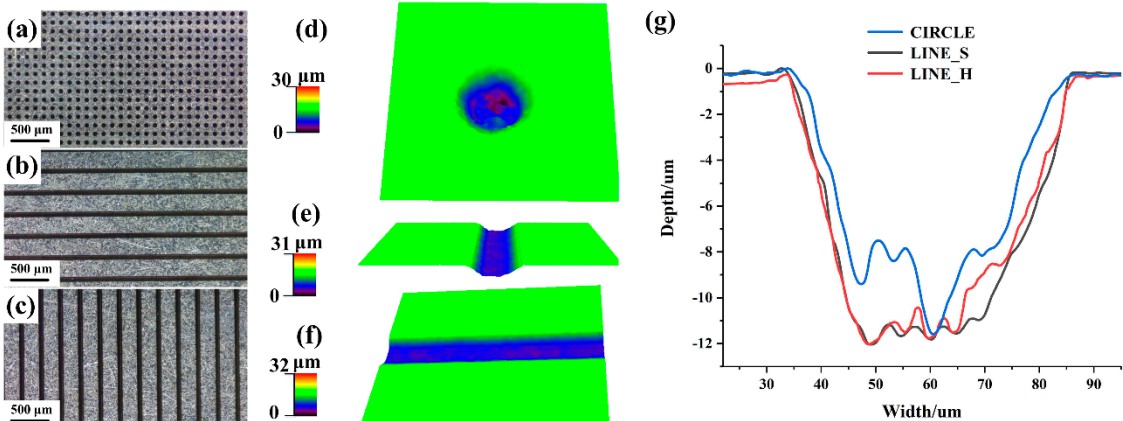

**Figure 3.** Surface morphologies of the micro-textures. (**a**) surface with micro-dimples, (**b**) surface with longitudinal micro-channels, (**c**) surface with lateral micro-channels, (**d**) micro-dimple, (**e**) longitudinal micro-channel, (**f**) lateral micro-channel, (**g**) comparison image of cross-section profiles. CIRCLE represents micro-dimple, LINE_S represents longitudinal micro-channels and LINE_H represents lateral micro-channels.

## 3.2. Effect of the Morphology of Micro-Texture on Adhesion Properties

To investigate the effect of different morphologies of micro-textures on adhesion properties, electrode 1, electrode 2, electrode 3 and electrode 4 (i.e., electrode without micro-texture, electrode with micro-dimples, electrode with longitudinal micro-channels, and electrode with lateral micro-channels) were performed in electric cutting experiments. Take the images of electrode surfaces after the first and fourth electric cutting experiments as the example, as shown in Figure 4. It can be seen that the tissue adhesion on electrode 1 is the most serious. The charring tissue covered almost the entire surface of electrode 1 after four electric cutting experiments. Compared with electrode 1, tissue adhesion is slightly less on electrode 2. The tissue adhesion on the electrode with longitudinal micro-channels reduces significantly, and tissue adhesion on the electrode with lateral micro-channels is least among the four kinds of electrodes.

In order to further explore the adhesion behavior of different electrodes, the relationship between tissue adhesion mass and cutting trial number was investigated, as shown in Figure 5. It can be seen that the adhesion mass of all electrodes increases with the increasing of cutting trial number. After the first electric cutting experiment, tissue adhesion mass of electrode 1 reaches 2.0 mg, which is most of all. The adhesion mass of electrode 2 and electrode 3 is 1.5 and 1.1 mg, respectively. Tissue adhered to electrode 4 is least, and the adhesion mass is only 0.9 mg. Compared with electrode 1, the tissue adhesion mass of electrode 2, electrode 3 and electrode 4 decreases by 25%, 45%, 55%, respectively. After the fourth cutting experiment, adhesion mass of electrode 1, electrode 2, electrode 3 and electrode 4 is 4.1, 3.4, 2.9, and 2.7 mg, respectively. The tissue adhesion mass of electrode 2, electrode 3 and electrode 4 decreases by 20%, 29% and 34%, respectively. It can be seen from the experimental results that micro-textures play a crucial role in improving the adhesion properties of electrodes. Among the textured electrodes, lateral micro-channels exhibit the most effective anti-adhesion property.

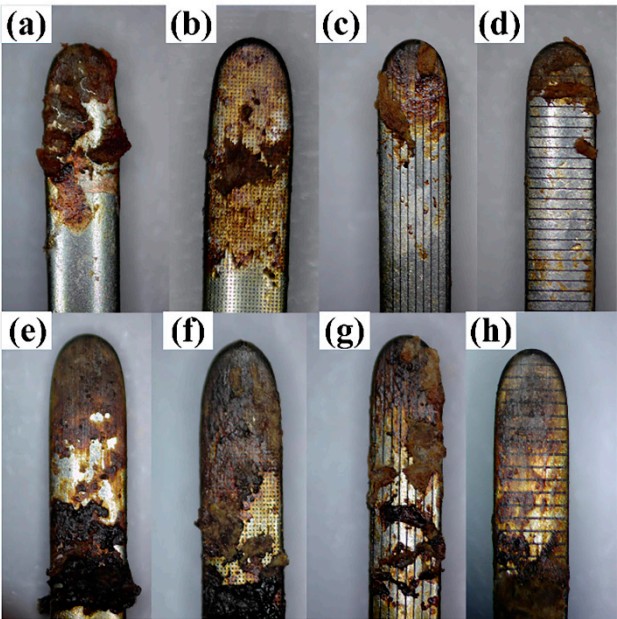

**Figure 4.** Adhesion comparison between electrodes with micro-textures of different morphologies. (**a–d**) represent the adhesions of electrode 1, electrode 2, electrode 3 and electrode 4 after the first cutting experiment, respectively. (**e–h**) represent the adhesions of the electrode 1, electrode 2, electrode 3 and electrode 4 after four cutting experiments, respectively.

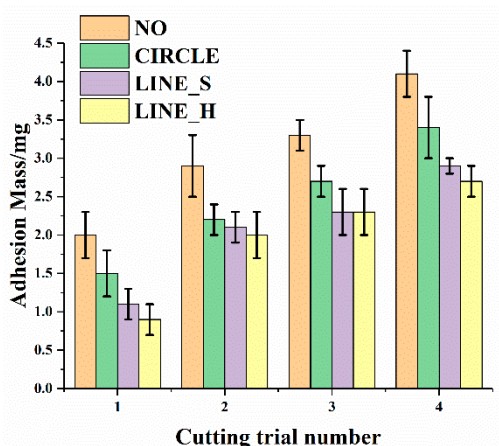

**Figure 5.** Comparison of tissue adhesion mass related to cutting times between electrodes with different morphologies.

### 3.3. Effect of Area Density on Adhesion Properties

Electrode 4, electrode 5, electrode 6 and electrode 7 (lateral micro-channels with area densities ranging from 10% to 40%) were used to investigate the effect of area density on adhesion properties. The images of electrode surfaces after electric cutting experiments are shown in Figure 6. It can be seen that the tissue adhesion on electrode 5 is more serious than others. The tissue adhesions on electrode 6 and electrode 7 are similar, which reduce significantly compared with the electrode without micro-texture.

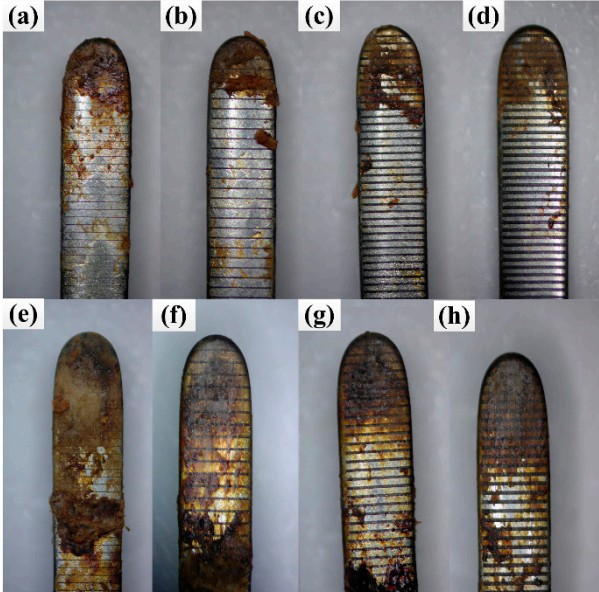

**Figure 6.** Adhesion comparison among micro-textures with different area densities. (**a**–**d**) represent the adhesions of electrode 5, electrode 4, electrode 6 and electrode 7 after the first cutting experiment, respectively. (**e**–**h**) represent the adhesions of electrode 5, electrode 4, electrode 6 and electrode 7 after four times cutting experiments, respectively.

The relationship between adhesion mass and area density was investigated, as shown in Figure 7. Repeated experiments were still performed. The experimental results show that the adhesion mass increases as the cutting times increase. After the first electric cutting experiment, the adhesion mass of electrode 5, electrode 4, electrode 6 and electrode 7 is 1.1, 0.9, 0.8, 0.6 mg, respectively. Compared with electrode 1, the adhesion mass decreases by 45%, 55%, 60% and 70%, respectively. After four electric cutting experiments, the adhesion mass of electrode 5 is the most among the four electrodes, which is 2.9 mg. The adhesion mass of electrode 6 and electrode 7 is similar, which is 1.9 and 1.7 mg, respectively. Compared with electrode 1, the adhesion mass of electrode 5, electrode 4, electrode 6 and electrode 7 decreases by 29%, 34%, 54%, 59%, respectively. Thus, it can be seen that the better anti-adhesion property can be obtained when the surface micro-textures with larger area density are created.

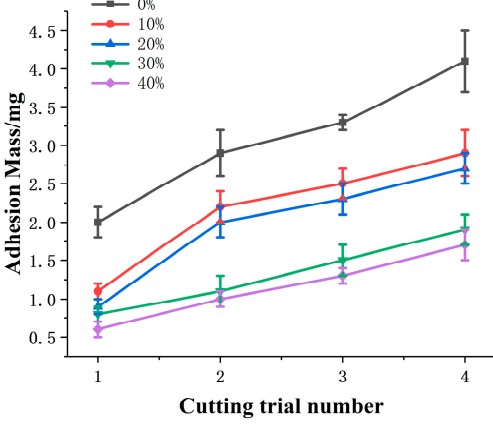

**Figure 7.** Comparison of tissue adhesion mass related to cutting times among electrodes with different area density.

### 3.4. Temperature Distribution

A finite element model was built to simulate the temperature distribution of the liver tissue in the electrical cutting process. Temperature distributions of the liver tissue for electrode 1, electrode 2, electrode 3 and electrode 4 at 1 and 3 s are shown in Figure 8. The finite element analysis results show that the area near the tip of the electrode can reach a higher temperature than the rest, thus causing more serious thermal damage and soft tissue adhesion. It can be seen from Figure 4a–d that the soft tissue adhesion on the electrode is mainly distributed near the curved area after the first electric cutting experiment, which is consistent with the simulation results. The temperature in the area of soft tissue which in contact with the electrode all increases with time, but the temperature in tissue cut by electrode 1 is higher than the others. The temperature in the soft tissue cut by electrode 1 is 82 °C at 1 s, which is the highest of all. The temperature in tissue cut by electrode 2 is 80 °C, and the temperature in tissue cut by electrode 3 is 75 °C. The temperature cut by electrode 4 is 73 °C, which is the lowest of all at 1 s. Compared with the others, the temperature of tissue cut by electrode 1 increases most rapidly, and the highest temperature reaches 116 °C at 3 s, as can be seen in Figure 8e. However, the lowest temperature in Figure 8h) is only 93 °C. Also, it can be seen that the temperature distribution is most concentrated in soft tissue cut by electrode 1. Compared with the temperature distribution in tissue cut by electrode 1, the center of the temperature field has a slightly dispersed trend in tissue cut by electrode 2. For tissue cut by electrode 3 and electrode 4, the temperature distributions are obviously dispersed, and the temperature in tissue cut by electrode 4 is lower than that of electrode 3.

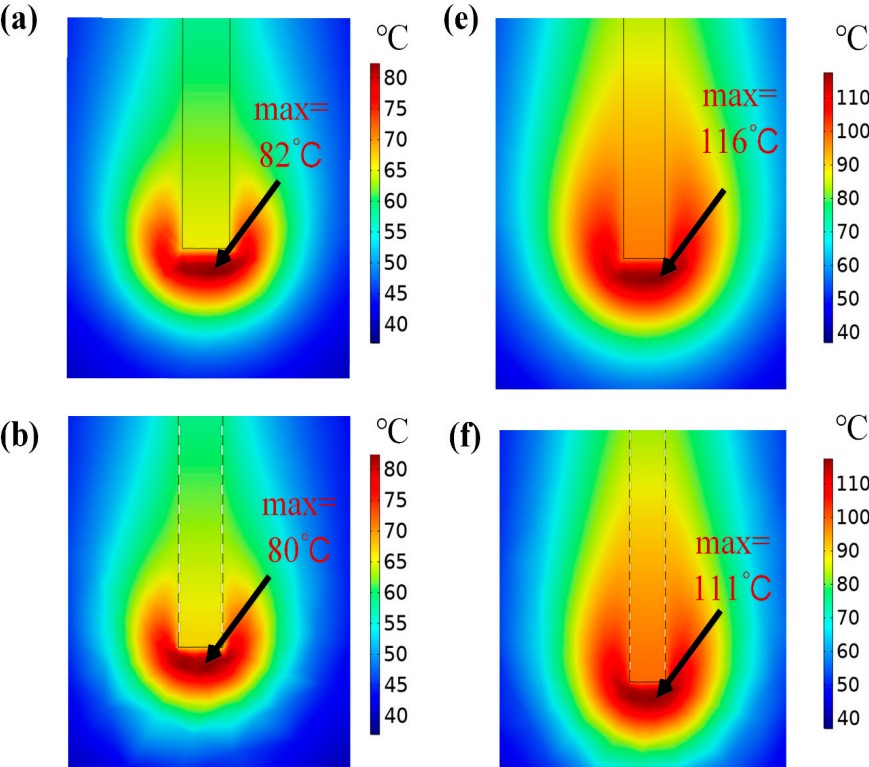

**Figure 8.** *Cont.*

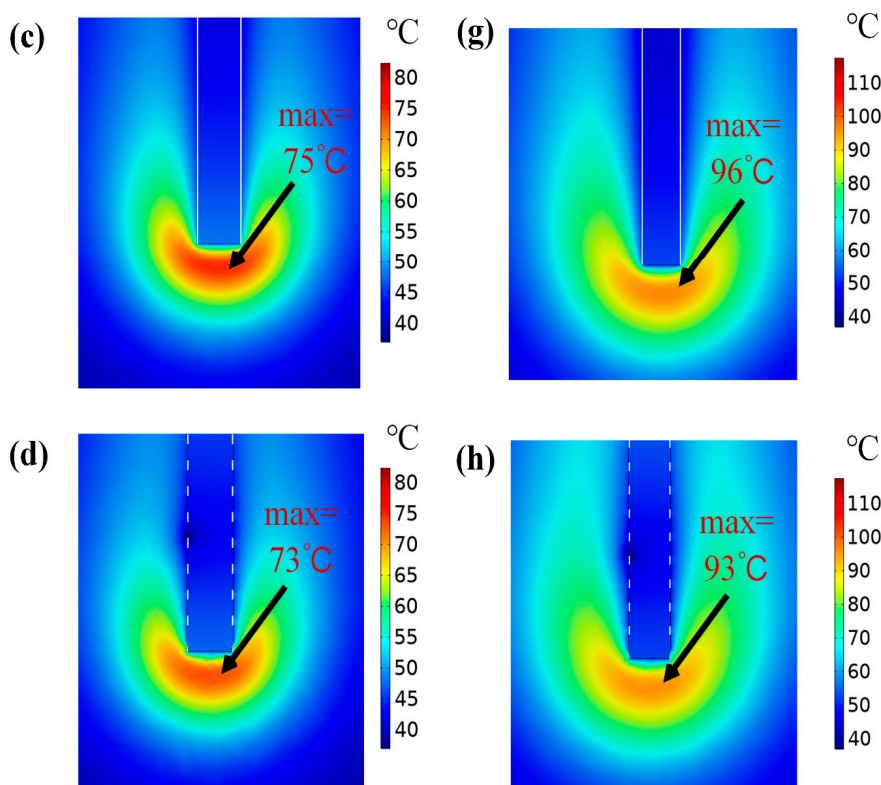

**Figure 8.** Cross-sectional view of temperature distribution. (**a**) temperature distribution in tissue cut by electrode without micro-texture at 1 s, (**b**) temperature distribution in tissue cut by electrode with micro-dimples at 1 s, (**c**) temperature distribution in tissue cut by electrode with vertical micro-channel at 1 s, (**d**) temperature distribution in tissue cut by electrode with horizontal micro-channel at 1 s. (**e**) temperature distribution in tissue cut by electrode without micro-texture at 3 s, (**f**) temperature distribution in tissue cut by electrode with micro-dimples at 3 s, (**g**) temperature distribution in tissue cut by electrode with vertical micro-channel at 3 s, (**h**) temperature distribution in tissue cut by electrode with horizontal micro-channel at 3 s.

### 3.5. Discussion

In this paper, the influence of morphology and area density of micro-textures on anti-adhesion properties was investigated. In general, the anti-adhesion properties are improved due to the fabrication of the micro-textures on the electrode, but affected by morphology and area density of micro-textures.

It has previously been shown that the adhesion between electrode and soft tissue is a physical adhesion caused by temperature [4,35]. The created micro-textures contain some entrapped air, which has a low heat conductivity coefficient, and acts as a thermal barrier for the transfer of heat to soft tissue, thus reducing the thermal damage to the soft tissue. Different distributions of micro-texture units on the textured electrode surfaces cause different segmentation effects on the temperature field during the electric cutting process. As can be seen in Figure 3, micro-dimples are distributed in a dispersed form on the electrode surface. The micro-channels are equivalent to dividing the electrode surface into multiple and discontinuous parts. The air distributions in the micro-channels are more concentrated, and the segmentation effect of micro-channels is more concentrated and obvious. Thus, micro-channels can achieve a more obvious segmentation effect on the temperature field than micro-dimples.

Furthermore, although the area densities of different micro-textures are the same, the surface morphology of electrode with different textures is different. The electrode surface with micro-dimples is large area of continuity, and the contact between soft tissue and electrode surface is also a large area of continuity during the electric cutting process. The micro-channels divide the electrode surface into some discontinuous parts. Therefore, the adhesion between the soft tissue and the electrode with

micro-dimples is continuous in a large area. The adhesion between the soft tissue and the electrode with micro-channels is based on each dispersed unit rather than a large area. Therefore, the tissue adhesion on the electrode with micro-channels is less than that with micro-dimples. The continuous tissue adhesion is less likely to produce when the more dispersed micro-channel units are created on the electrode surface. The lateral micro-channels divide the electrode surface into more dispersed parts than longitudinal micro-channels. Therefore, there is the least soft tissue adhesion on the electrode with lateral micro-channels. The order of soft tissue adhesion from serious to slight is the electrode without micro-texture, electrode with micro-dimples, electrode with longitudinal micro-channels, and electrode with lateral micro-channels.

The influence of area density on adhesion behavior was investigated by using lateral micro-channels. With the increasing of the area density, the contact area between electrode and soft tissue decreases due to the increasing width of the micro-textures. Thus, the thermal damage and tissue adhesion is much less. Besides, it can be seen from the experimental results that the adhesion mass of the textured electrodes with area densities of 30% and 40% is similar, which indicates appropriate parameters of the micro-textures for improving anti-adhesion properties are obtained. Therefore, the anti-adhesion property of micro-textures with larger area densities is not studied further in this paper.

## 4. Conclusions

In this research, three kinds of micro-texture with distinct morphologies were fabricated on the electrode surface by LST. Electric cutting experiments were performed to investigate the adhesion properties of the textured electrodes. Moreover, the anti-adhesion mechanism of different textured surfaces was verified by COMSOL finite element analysis software. The following conclusions were summarized:

- Micro-textures with different morphologies and area densities were fabricated on the electrode surfaces. The micro-textures improved the adhesion properties of the electrode due to the reduction of contact area between the electrode and soft tissue, and the effect of air in the micro-textures.
- Different morphologies of micro-textures lead to different adhesion behavior of electrodes. The order of adhesion mass from more to less is the electrode without micro-texture, the electrode with micro-dimples, longitudinal micro-channels, and lateral micro-channels. Furthermore, the influence of micro-texture with different morphologies on temperature and adhesion behavior in the cutting process was verified by COMSOL multiphysics simulation software. The simulation results showed that different micro-textures can segment the temperature field in different ways, and reduce the transfer of heat to different degrees, thereby reducing soft tissue adhesion.
- Different area densities of micro-textures can lead to different effects on adhesion behavior of the electrode. With the increase of area density, adhesion mass decreased clearly. When the area density increased to a larger value, adhesion mass no longer decreased significantly. The better anti-adhesion effect can be obtained by combining lateral micro-channels with a large area density.

**Author Contributions:** Investigation, methodology, software, writing—original draft preparation, C.Z.; Investigation, J.L.; supervision, conceptualization, writing—review and editing, X.W. All authors have read and agreed to the published version of the manuscript.

**Funding:** This research was funded by National Natural Science Foundation of China (51705258), the Fundamental Research Funds for the Central Universities (KJQN201843), National Undergraduate Innovation and Entrepreneurship Training Program (201910307076Z).

**Conflicts of Interest:** The authors declare no conflict of interest.

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
