# Peer review of "Adhesion Behavior of Textured Electrosurgical Electrode in an Electric Cutting Process"

_coatings, doi:10.3390/coatings10060596_

Round 1

Reviewer 1 Report

This article presents a method based on laser texturing for reducing adhesion of soft tissue on electrosurgical electrodes for an electric cutting process. Three different surface morphologies are explored: micro-dimples, horizontal channels and vertical channels. The article is generally well written. The literature review in the introduction is adequate. The experimental procedure is clear, logical and easy to follow, including the measurement of tissue adhesion mass after cutting at different speeds and then with different pattern area densities.

I have a few suggestions:

- Please describe the attenuation system used to control the laser pulse energy. This is included in Figure 1 but is not referenced within the text. 

- Differentiation between “horizontal” and “vertical” micro-channels is not immediately clear to the reader, as this might depend on the electrode orientation. One initially has to check the electrode number, then check Table 1 to see the notation, then look at the photo to understand the orientation. In alternative, the terms “lateral” and “longitudinal” could be employed, or “horizontal (lateral)” and “vertical (longitudinal)”.

- Within the discussion the authors talk about air in micro-textures; however, when discussing the thermal simulation, what really seems to the be the driving factor is a reduction in contact area between the electrode and tissue, and therefore a reduction in temperature increase. It would seem more logical to discuss the effects of laser texturing on the contact area rather than the volume of air, or at least separate the two concepts more clearly. For example, much deeper dimples could be created to achieve the same volume as the micro channels, but do the authors think this would significantly change the outcomes? 

- Figure 8 presents a very macroscopic view of the temperature distribution. It would be useful to present some images showing the local temperature variation on the same scale as the texturing to see what the physical effect might be.

- There are minor English language errors throughout the paper that should be addressed.

Reviewer 2 Report

This manuscript describes the modification of an electrosurgical electroblade in an effort to minimize tissue adherence as a result of localized heating. This is an interesting area for the use of modified surface structure, and for the most part the work is well done and interesting. It should be of interest to the readers of the journal and has the potential to have an impact on the field.

That said, there are several areas of the manuscript that will need to be improved.

The COMSOL experiments produce data. The methods for them should be put into the experimental section, and the results should go in the results section. That will help fix one of the other comments regarding the discussion of air and temp noted below.

Please clarify if table 1 is the desired characteristics or the actual measured ones. The format suggests its desired, but the sentence immediately before it is about the measurements so its unclear.

What was the source of the porcine liver? If it came from a slaughterhouse, that should be noted. If it came from an animal sacrificed for these studies, then that should be noted along with the animal approval statement.

The x-axis on figure 5 should be renamed. I understand the authors mean the number of times they cut the tissue, but the phrase cutting time sounds like the time it took to complete a cut. Perhaps cutting trial number, or simply cut number would clarify this. The same change should happen in figure 7.

The discussion of adhesion masses following figure 7 lists only a mean value. There should be an error associated with these measures (either a standard deviation or standard error) that should be listed here to show variability in the measures.

The beginning of the discussion talks about heat measures that are not presented yet. Moving the COMSOL simulations into a results section would allow the authors to discuss this section with the reader being aware of the data they are referring to.

In the discussion about the entrapped air the authors state that there is more air in each microchannel unit than in the dimples. I think the authors mean an individual microchannel vs an individual dimple, but if the surface area coverage for each of the textures is the same and the depth is the same, the overall total amount of air entrapped is the same. In fact, the whole discussion of air and segmentation effects on page 8 in the discussion is very confusing and should be rewritten.

The second to last sentence of the conclusion is not necessarily supported. The authors go to 40% surface coverage, and suggest that since there was no change from 30 to 40, then the amount of adherent tissue is saturated. It is not clear that statement can be made from 2 points that “are similar” without any sort of statistical test and demonstrating that going to 50% does not result in a further improvement (even if “similar”). Similarly, for the same sort of reasons, it’s not clear that they have found the “ideal” anti-adhesion effect.

Reviewer 3 Report

In this work the adhesion behavior of textured electrosurgical electrode in electric cutting process was carry out. The paper is interesting, attractive topic for investigation and can be publish in Coatings.

I think the paper can be accepted for publication as it stands.

Reviewer 4 Report

Dears,

The experimental results are interesting scientifically sound. Despite of these interesting results, the paper should be slightly corrected.

  • English should be improved
  • in the Introduction part beside with LST method, please mention also LIPSS method that can significantly improve adhesion and wetting properties. 

Have a look paper:

"Laser nanopatterning for wettability applications", Journal of Micro-and Nano-Manufacturing, 5, 021008 (2017).

Finally, the numbers on Figure 8 are not readable, please change the color and size.
